# Stability and Anti-Aging of Encapsulated Ferulic Acid in Phosphorylated Rice Starch

**DOI:** 10.3390/molecules27113463

**Published:** 2022-05-27

**Authors:** Jittraporn Pueknang, Nisakorn Saewan

**Affiliations:** 1School of Cosmetic Science, Mae Fah Luang University, 333, Moo.1, Thasud, Muang, Chiang Rai 57100, Thailand; jittraporn.pue12@lamduan.mfu.ac.th; 2Cosmetic and Beauty Innovations for Sustainable Development (CBIS) Research Group, Mae Fah Luang University, 333, Moo.1, Thasud, Muang, Chiang Rai 57100, Thailand

**Keywords:** phosphorylated starches, clinical evaluation, ferulic acid, skin lightening, anti-aging

## Abstract

Ferulic acid (FA) provides broad biological functions that have been used in cosmetics formulation as a photoprotection, anti-aging, and brightening agent. However, its application is limited by its tendency to deteriorate by exposure to heat, humidity, and light. This study aimed to enhance the stability of FA by encapsulation in phosphorylated rice starch (PRS) and evaluate its effect on improving human skin. First, FA was encapsulated in PRS and characterized by FTIR, SEM, XRD, and DSC. Then, its stability when exposed to a temperature of 45 °C and light and its anti-aging effect on 16 volunteers were investigated. The results indicated that FA was successfully encapsulated in PRS with an encapsulation yield of 77%, EE (73%) and LE (65%). After 1 month at the high temperature/80%RH, the encapsulated FA retained its quantity (70%), whereas free FA was retained at only 50%. Under light exposure conditions, the encapsulated FA was retained at 65%, which was higher than FA (35%). Franz diffusion cell was used and demonstrated that PRS provided the controlled release of FA. Application of encapsulated FA and FA creams showed an absence of skin irritation in all volunteers. After 1 month, the encapsulated FA cream was found to be better than the FA cream on skin lightening, elasticity, smoothness, roughness, scaliness, and wrinkle. The results indicated that PRS is a potential wall material for enhancing the stability of FA, resulting in more efficacious skin lightening and anti-aging properties.

## 1. Introduction

Ferulic acid (FA), 4–hydroxy–3–methoxycinnamic acid, is a phenolic compound, which provides photoprotection, antioxidant, anti-tyrosinase, and antimicrobial properties [1,2,3,4]. It has strong free radical scavenging activity and absorbs UV light. It can be used as a multifunctional agent in cosmetic products, such as anti-aging, anti-wrinkle, skin lightening, anti-pollution, and photoprotection. However, its application in cosmetic products has been limited due to it undergoing high thermal relative humidity (RH > 76%) and light-induced decomposition through decarboxylation into 4–vinyl guaiacol (4–hydroxy–3–methoxystyrene) and other derivatives, which cause a change in product color and a reduction in its efficacy [5]. In recent years, several attempts have been made to encapsulate FA with polysaccharides to enhance its stability. Currently, ferulic acid has been successfully encapsulated onto different polysaccharides, such as cyclodextrin, chitosan, and starch [6,7,8]. It has been shown that the encapsulation of ferulic acid in polysaccharides enhances its stability and broadens its application in many fields. Phosphorylated starches have proved to be a good alternative for use as a matrix in controlled active delivery systems and enhance the stability of the phenolic compound [9,10,11,12,13]. Moreover, García-Gurrola et al. have reported that phosphorylated starch had high thermal stability when used as a microcapsule of sorghum phenolic compounds, with great resistance to heat treatments while maintaining structural integrity [11]. Therefore, phosphorylated starch is a sound choice to use as an encapsulating agent due to its affordability, absence of waste products, and short preparation time. In this study, the stability and efficacy of encapsulated FA in PRS were investigated for the first time. The chemical and physical characterization of the encapsulated FA were investigated by several instrumental methods, and its stability was further evaluated by storage under high temperature/80% RH and exposure to light for 1 month. The effect on skin color was evaluated by using a Mexameter; skin firmness, which included gross elasticity (R2), net elasticity (R5), and portions of viscoelasticity (R7), was assessed by a Cutometer; and skin smoothness (SEsm), roughness (SEr), scaliness (SEsc), and wrinkles (SEw) were analyzed by Visioscan.

## 2. Results

### 2.1. Preparation of Encapsulated FA

The ratio of wall material and core substance is an important key in the preparation of encapsulation in reaching the appropriate application. As such, the concentration of FA was studied by fixing the molar number of phosphorylated rice starch (PRS) and varying the FA from 1.0 to 3.0 molars. PRS appeared as a light-yellow granule, while FA presented a yellow homogeneous powder. After encapsulation, the loaded FA was observed as a yellowish powder, with yields ranging between 40 to 77%. The higher yield was obtained when the molar of FA was 2.0 and higher.

The encapsulation efficiency (EE) and loading efficiency (LE) were summarized in Table 1. Data showed that with an increase in the molarity of FA, the EE also increased and reached a maximum of 73.10%. Similarly, LE tended to increase with an increase in the molarity of FA and FA content ranging from 21.31 to 65.40%. By increasing the molarity of FA from 2.0 up to 3.0, the EE and LE did not significantly (*p* > 0.05) increase. This could be due to reaching the saturation solubility of FA in PRS, which partially limits its incorporation into the wall material. Other researchers also observed a decrease in EE and LE when increasing the proportion of the core material [14,15]. Thus, the optimal molar ratio of PRS and FA is 1.0:2.0, and this ratio was chosen for further study.

### 2.2. Scanning Electron Microscope (SEM)

The morphological characterization of PRS, FA, physical mixture, and encapsulated FA was investigated by using SEM (Figure 1). PRS granules clumped together and irregular granules displayed a rough surface. FA was observed as a needle-like crystal. The physical mixture showed both characteristic FA and PRS, while the encapsulated FA completely lost the granular structure of FA, presenting large amorphous aggregates, which is consistent with the XRD results. The sizes and shapes of the complex were different from those of the starting PRS and FA. This suggested that the complex was structurally districted from the starting materials. These results were also observed by Wang, who reported that the inclusion of complex particles showed irregular pieces of amorphous aggregates and loss of the original morphologies [16].

### 2.3. Fourier Transform Infrared Spectroscopy (FT-IR)

The interaction between PRS and FA was investigated by using FT-IR in Figure 2. The PRS spectrum presented the peaks around 3600–3000 cm^−1^ indicating the stretching vibration of O–H bonds. The characteristic absorption peak of PRS at 1214 cm^−1^ was assigned to a symmetrical stretching of P=O, a band at 1129 cm^−1^ related to symmetrical and asymmetrical stretching of the PO_2_ group, and a band at 889 cm^−1^ related to the P–O bond. The IR spectrum of FA showed broad characteristic peaks at 3436 cm^−1^, which was assigned to the hydroxyl group, a band at 1691 cm^−1^ related to the stretching vibration of carbonyl, and bands at 1620, 1515, and 1431 cm^−1^ were associated with C=C and aromatic skeleton vibrations. Additionally, the sharp bonds at 850 and 804 cm^−1^ were due to the two adjacent hydrogen atoms on the phenyl ring of the FA, which was compatible with the previous study [8,15,16]. The physical mixture spectra presented peaks more similar to FA, which indicated that the ratio of the FA was higher than the PRS (2:1), and small peaks of PRS can be observed with no interaction with FA.

The spectrum of the encapsulated FA showed the broad peak at around 3436 cm^−1^, which decreased in intensity due to the decrease in the number of hydroxyl groups, as a complex formed between the PRS and FA. The peak of interaction between the FA and PRS emerged at 1688 cm^−1^. Moreover, the bands of the aromatic nucleus shifted and were diminished to 1620, 1518, 1433 cm^−1^, whereas the bands at 853 and 803 cm^−1^ reduced in their intensity.

### 2.4. X-ray Diffraction (XRD)

The X-ray diffractograms of PRS, FA, encapsulated FA, and physical mixture are shown in Figure 3, which confirms changes in the crystalline phase. PRS demonstrated peaks at 2θ of 13° and 20°, characteristic of a V-type crystalline structure. In contrast, the physical mixture observed a crystalline structure as both starting materials. As compared with PRS, the encapsulated FA exhibited the same broader peak at 13° and 20°, but it did not exhibit the crystalline nature peaks of the FA, which showed strong reflection (2θ) at 9°, 10.4°, 12.8°, 15.6°, 17.4°, 18.1°, 25.0°, and 26.5°, indicating the formation of a significant amount of amorphous material. This result showed that FA lost its crystallinity after encapsulation, which was consistent with the study of Mathew and Abraham [8] and Wang et al. [16].

### 2.5. Differential Scanning Colorimetry (DSC)

The DSC thermograms and data of samples are shown in Figure 4 and Table 2. The DSC curve of PRS showed a broad endothermic peak between 36 and 120 °C for dehydration and decomposition due to phase transition of the ordered structure to the random state. In terms of the FA, it presented a sharp endothermic melting point at 177.53 °C, which was the decomposition of a crystalline state. The case of the physical mixture showed both individual characteristic peaks of the PRS and FA. Contrary to the encapsulated FA, the crystal melting peak of free FA vanished, indicating that after loading into the matrix of PRS, the FA lost its crystalline state and shifted to a broad endothermic peak as PRS (37–142 °C), which exhibited the interaction between FA and PRS. The disappearance of the endothermic peak of FA has been reported when it was completely loaded in wall material [15,16]. The DSC result was in correspondence with the XRD diffractogram, which did not observe the crystalline peak in the encapsulated FA.

### 2.6. Particle Size and Zeta Potential Measurement

Particle size, PDI, and zeta potential of the PRS and encapsulated FA are presented in Table 3. PRS had a particle size of about 71.51 ± 7.01 nm, with a PDI of 0.297 ± 0.007, and zeta potential was −67.72 ± 3.18 mV. After encapsulation, the encapsulated FA showed an increase in particle size to 73.90 ± 2.67 nm and PDI to 0.394 ± 0.005, while a decrease in zeta potential to −14.85 ± 1.90 mV was observed. This might be due to the aggregated granules after encapsulation, revealing a low electrostatic repulsion and high van der Waals forces, as has been observed earlier by Das and Wong [17].

### 2.7. Accelerated Storage Stability Test

The exposure to extremely heat, humidity and light plays an important role in terms of the FA stability, which is directly responsible for its effectiveness. To ensure the quality of the encapsulated FA for use in cosmetic products, the effect of high temperature and humidity in accelerating any change of material was evaluated at 45 °C/80% RH. According to the ICH guideline on photostability (ICH-Q1B), the procedure to evaluate photostability of a compound can be tested with fluorescent lamps or UV lamps. In addition, during storage, product display, and consumer use, cosmetic products are more exposed to fluorescent light than UV light environments. Thus, to evaluate the effect of light on the stability of the FA and encapsulated FA, the samples were explored under fluorescent light conditions for 1 month (Figure 5).

After one-day heat treatment, 96% of the encapsulated FA was retained, which was higher than for the free FA (about 65%). During a period of 1 to 15 days, the encapsulated FA slightly decreased from 96 to 72%, whereas the FA decreased from 65 to 50%. At the end of the treatment, the encapsulated FA was retained at more than 70%, whereas the FA was retained at about 50%. The effect of light on the stability of the FA was determined by storing the test samples under fluorescent light. The results showed that 98% of the encapsulated FA was retained after 1 day, whereas the FA was retained at about 67%. Between 1 to 15 days, the encapsulated FA slightly decreased from 98% to 72%, whereas the FA decreased from 67% to 34%. At the end of the experiment, the encapsulated FA retained more than 65%, whereas the FA retained only about 35% (Figure 5a).

Thus, the stability of the encapsulated FA under both heat/80% RH and light retained about two-fold more than the free FA (Figure 5b). Significant differences (*p* < 0.05) were observed among the free and the encapsulated FA. This evidence showed that encapsulation in the PRS as wall material offered protection to the FA under heat/80% RH and light treatment. Additionally, encapsulation can act as barrier layers, limiting the diffusion of oxygen and moisture, preventing the penetration of light, and reducing the sensitivity to sharp thermal fluctuations. Thus, encapsulation in PRS offers a longer product shelf life during storage under exposure to high temperatures/humidity and light.

### 2.8. Penetration Experiment

This study is crucial to the release of the FA from the encapsulated FA and penetrates through the stratum corneum; thus, the active can act at the site of action (inner epidermis and dermis). Here, cellulose acetate membranes with 0.45 μm pore size were used as a barrier and release medium. The release of FA followed by permeation through the membrane to the receptor fluid was evaluated using the Franz diffusion cell. The release of FA and encapsulated FA (mg/mL) was plotted against time. Figure 6 indicates that the release rate of FA was rapid for the initial 10 h and then slowed. Interestingly, the encapsulated FA release rate during the first 10 h was slower than the FA’s. However, the active release from the FA and encapsulated FA after 10 h continuously increased. Overall, using PRS as wall material controlled the release of FA. Another researcher has also reported a slow release of the encapsulated compound from crosslinked starch [10].

### 2.9. Formulation

After incorporating the free FA and encapsulated FA as anti-aging ingredients in an emulsion cream base, the FA product appeared as pale yellow, whereas the encapsulated FA cream appeared as orange yellow, as can be seen in Figure 7.

The pH value of the FA (5.04) and the encapsulated FA (5.43) creams is slightly acidic, which is suitable for maintaining a good skin barrier, moisturizers, and bacteria protection. The encapsulated FA cream (7440 ± 72.11 cp) had more viscosity than the FA cream (4366 ± 30.55 cp) due to the contained rice starches acting as a thickening agent in the formulation. Overall, both creams were satisfactorily acceptable for topical application with appreciable viscosity no running down from the application area, good spreadability with thixotropic property, and non-stickiness.

### 2.10. Clinical Study

#### 2.10.1. Skin Irritation Testing

The possibility of skin irritation from both creams was evaluated by using a closed patch test. Sodium lauryl sulfate (SLS) and water were used as positive and negative standards, respectively. Skin irritation was evaluated at 30 min and 24 h after removing the patch, as per the scores reported in Table 4, which describe the severity of erythema, oedema, or other skin irritations. SLS irritation and side effects were observed in all volunteers, and the mean irritation index (M.I.I.) of this surfactant was 1.12, while the M.I.I. of both the FA and encapsulated FA creams was 0; therefore, they were classified as non-irritant products.

#### 2.10.2. Skin Lightening Effect

FA is used in the skin lightening formulations because it inhibits tyrosinase activity in melanogenesis and inhibits melanocytic proliferation, which plays an important role in human skin color [18]. To evaluate the skin lightening efficacy of the encapsulated FA in comparison to free FA, a single-blind test was conducted with 16 subjects. The volunteers were instructed to apply the test products on both the left and right upper arms randomly for each person for 4 weeks. The average age of the subjects was 41 ± 5.7 years; the oldest participant was 55, and the youngest was 36. The results showed that the melanin content of volunteers who applied the encapsulated FA and free FA cream continuously declined. After 2 weeks, the results demonstrated that the change in melanin content was significantly lower than the baseline value at the start of the test by 5.74 and 3.86% for the encapsulated FA and free FA cream, respectively (*p* < 0.05). At week 4, the change in melanin content continuously decreased from the baseline by 11.43 and 5.69% for the encapsulated FA and FA cream, respectively (*p* < 0.05) (Figure 8).

Furthermore, significant differences (*p* < 0.05) in lightening efficacy were observed between the encapsulated FA and free FA creams after 2 and 4 weeks (Figure 9).

#### 2.10.3. Anti-Aging Effect

The results are displayed in a curve from which a series of interesting R parameters can be calculated. This study investigated the gross elasticity of the skin (R2), which is resistant to mechanical force versus to its returning, the net elasticity (R5) elastic portion of the suction part versus the elastic portion of the relaxation part, and the portion of the elasticity (R7) compared to the complete curve. The skin showed more elasticity the closer its R value was to 1 (100%). After 2 weeks, the 3R of skin elasticity of volunteers who applied the encapsulated FA cream was greater than that of the FA cream (*p* < 0.05). The change of skin elasticity significantly increased from the baseline by 5.73 and 3.88% (R2), 6.12 and 3.36% (R5), and 6.87 and 3.88% (R7) for the encapsulated FA and FA cream, respectively. At 4 weeks, the 3R of skin elasticity of volunteers who applied the encapsulated FA cream was greater than that of the FA cream. The change of skin elasticity significantly increased from the baseline by 11.29 and 7.81% (R2), 13.84 and 6.96% (R5), and 14.46 and 7.22% (R7) for the encapsulated FA and FA creams, respectively (Figure 10, Figure 11 and Figure 12) In addition, significant differences (*p* < 0.05) in anti-aging efficacy were observed between the encapsulated FA and free FA creams (Figure 12, Figure 13, Figure 14 and Figure 15).

#### 2.10.4. Visioscan

Visioscan provides a unique description of the skin topography directly from the skin using a special UV–A light video camera with high resolution. It was developed specifically to study the skin surface directly. The images show the skin’s structure and its level of dryness. Visioscan can analyze the gray level distribution and skin smoothness (SEsm), skin roughness (SEr), scaliness (SEsc), and wrinkles (SEw).

After 2 weeks, the change of skin smoothness increased from the baseline by 22.89 and 22.25%, with reduced skin roughness (14.45, 17.10%), scaliness (20.56, 14.47%), and wrinkle (22.47, 26.33%) for the encapsulated FA and FA creams, respectively. In addition, significant differences (*p* < 0.05) in scaliness were observed between the encapsulated FA and FA creams despite there being no significant differences in other SE parameters (Figure 16).

After 4 weeks, the SE parameters of the skin images of volunteers who applied the encapsulated FA cream were greater than those of the FA cream. The change of skin smoothness increased from the baseline by 45.69 and 43.79%, with reduced skin roughness (27.6, 23.59%), scaliness (50.50, 35.33%), and wrinkle (47.81, 35.03%) for the encapsulated FA and FA creams, respectively. In addition, significant differences (*p* < 0.05) in the reduction in scaliness and wrinkle were observed between the encapsulated FA and FA creams. The treated skin was photographed using Visioscan^®®^ VC 98 and compared with the photograph taken before treatment. The results demonstrated that both the encapsulated FA and FA cream reduced skin wrinkle, scaliness, and roughness after treatment (Figure 17). This is because the degradation of FA in the formulation affected product efficiency over time. Furthermore, the results indicated that PRS can be used as a wall material to enhance FA stability and efficiency in cosmetic products.

## 3. Materials and Methods

### 3.1. Reagents and Chemicals

Rice starch was purchased from Chang fun feung Company, Thailand. Ferulic acid from rice bran was obtained from Tsuno Rice Fine Chemicals Co., Ltd., Wakayama, Japan. Sodium tripolyphosphate (STPP) was purchased from Sigma-Aldrich (St. Louis, MO, USA). Sodium sulfate, sodium hydroxide, and ethanol were purchased from ACl Labscan (Bangkok, Thailand). Hydrochloric acid and hydrogen peroxide were acquired from Merck (Darmstadt, Germany). All reagents were of analytical grade.

### 3.2. Preparation of Encapsulated FA

PRS (1.25 g) was dispersed in deionized water (100 mL) at 90 °C for 30 min. Then, 2.75 g of ascorbic acid was dispersed in 30 mL of 1.0 M H_2_O_2_ and added with constant stirring for 30 min. Then, the reaction was cooled down to 30 °C before adding 50 mL of FA in ethanol solution and constantly stirring at room temperature for 24 h. After that, the sample was centrifuged at 9000 rpm for 10 min, and the supernatant was decanted. The precipitate was washed 3 times with ethanol (50 mL) and centrifuged at 9000 rpm for 10 min. Finally, the precipitate was dried by using a freeze dryer for 24 h. The encapsulated FA was stored at room temperature and kept in darkness for further analysis. The percent yield of the encapsulated FA was calculated by using the following equation.
(1)$$\%\;\mathrm{yield}=\frac{\mathrm{obtained}\;\mathrm{weight}\;\mathrm{of}\;\mathrm{encapsulated}\;\mathrm{FA}}{\mathrm{initial}\;\mathrm{weight}\;\mathrm{of}\;\mathrm{PRS}+\mathrm{FA}}\;\times \;100$$

In some analyses, the physical mixture was used to confirm the success of the encapsulated FA in PRS. The physical mixture was also prepared by mixing PRS and FA at the same ratio chosen from the study of optimal ratio (1:2 molar ratio) in ceramic mortars to obtain a homogeneous mixture for use as an analytical comparison.

In order to calculate the encapsulation efficiency (EE) and loading efficiency (LE), 10 mg of encapsulated FA was dissolved in 10 mL of DMSO and sonicated for 10 min. The solution was centrifuged at 9000 rpm for 10 min. The supernatant was collected, and its absorbance was measured at 315 nm, and the FA concentration was calculated by using a standard curve at 0.001–1.00 mg/mL. Finally, the EE and LE expressed as a percent were calculated according to the following equations.
(2)$$\mathrm{EE}\;(\%)=\frac{\mathrm{Weight}\;\mathrm{of}\;\mathrm{free}\;\mathrm{FA}\;\mathrm{in}\;\mathrm{encapsulated}}{\mathrm{Weight}\;\mathrm{of}\;\mathrm{initial}\;\mathrm{FA}\;\mathrm{added}}\times 100\;$$
(3)$$\mathrm{LE}\;(\%)=\frac{\mathrm{Weight}\;\mathrm{of}\;\mathrm{free}\;\mathrm{FA}\;\mathrm{in}\;\mathrm{encapsulated}}{\mathrm{Weight}\;\mathrm{of}\;\mathrm{encapsulated}\;\mathrm{FA}}\times 100\;$$

### 3.3. Scanning Electron Microscope (SEM)

The morphology of the samples was investigated using a scanning electron microscope (SEM) (LEO, 1450 VP LEO). The samples were stubs, coated for 60 s with gold–palladium using a sputter coater and photographed with a microscope at an accelerating voltage of 20 kV.

### 3.4. Fourier Transform Infrared Spectroscopy (FT-IR)

The FT-IR spectra of the samples were recorded using the FI-IR spectrometer (Perkin Elmer Spectrum100, PerkinElmer, Waltham, MA, USA). Samples were prepared by grinding the fine powder with KBr. The spectrum recorded the wave number at a resolution of 4 cm^−1^ from 4000 to 400 cm^−1^ with 32 scans.

### 3.5. X-ray Diffraction (XRD)

The crystalline structure of the samples was analyzed by using an X-ray diffractometer (PANalytical, X’Pert Pro MPD) with Cu–K radiation (λ = 1.54056 A)° at a target voltage and current of 40 kV and 30 mA. The scanning range and rate were 5–30° (θ) and 1.0°/min.

### 3.6. Differential Scanning Colorimetry (DSC)

Thermal analysis of the samples was performed by DSC apparatus (Mettler—Toledo, Switzerland). Each sample was placed in an aluminum pan and heated at the range of 10–220 °C (10 °C/min) under a nitrogen purge (50 mL/min). An empty aluminum pan was used as the reference.

### 3.7. Particle Size, PDI, and Zeta Potential Measurement

The sample particle size, polydispersity index (PDI), and its zeta potential were measured by Zetasizer (Malvern Worcestershire, UK; Nano S). For measuring the particle size, the samples (0.01%, *w*/*v*) were suspended in water and sonicated for 30 min at 40 KHz to completely disperse the particles.

### 3.8. Accelerated Storage Stability Test

In order to determine the effect of heat on the stability of the encapsulated FA and FA, the method was prepared following the previous study of Li et al., 2016 [14]. Briefly, 10 mg of samples was weighed in a small glass vial. The prepared samples were applied in a hot air oven (HCP108230V, Memmert) at 45 °C/80% RH (relative humidity) in the absence of light for 30 days. To study the exposure to light, the prepared samples were treated under a fluorescent light (Lamtan, 36W) at room temperature for 30 days. The distance between the bottom of the vial and the fluorescent light was 60 cm. After treating for a predetermined time at durations of 1, 3, 6, 9, 12, 15, and 30 days, the stored samples were dissolved in 10 mL DMSO and sonicated for 10 min. Then, the samples were centrifuged at 9000 rpm for 10 min, and the absorption at 315 nm of the supernatant was determined on a UV–vis spectrophotometer. The absorption of samples with and without treatment was recorded as to the initial and remaining amounts of FA in the samples. All storage samples were performed in triplicate, and the results were expressed as means and standard deviation.

### 3.9. Permeation Experiment

The permeation experiment was carried out according to the method described by Mattiasson [19]. Cellulose acetate membranes were washed to remove any dirt that could interfere with the execution of experiments and impregnation. Washing was performed by soaking the membranes in phosphate-buffered saline (PBS), pH 7.4, for 60 min prior to use. The membrane was mounted on a receiver cell with the stratum corneum side facing upward toward the donor cell. During the experiment, the receiver cells were filled with PBS (pH 7.4), which was continuously stirred at the rate of 300 rpm/min. Temperature (37 ± 0.2 °C) was maintained with an external circulating water bath. In the donor cell, 1 mL of 1 mg/mL FA solutions and encapsulated FA were separately applied homogeneously on the membrane. At pre-set time points (0.5, 1, 1.5, 2, 2.5, 3, 3.5, 4, 4.5, 5, 5.5, 6, 24, 30, 48, 54, and 72 h), 1 mL of the receptor solution was withdrawn, and the same volume of fresh receptor was added to the receptor cell in order to maintain the condition. Samples were analyzed for the amount of FA, which was quantified spectrophotometrically at 315 nm, using the FA standard curve (0.001–1.00 mg/mL).

### 3.10. Formulation

The FA and encapsulated FA creams were prepared according to the base cream reported in Table 5. Parts A and B were heated to 70 °C, and then, part A was added to part B. The stirring of part AB continued until well mixed. The mixture was homogenized for 15 min and cooled down to 50 °C; after that, part C was added to part AB. Next, the base cream was separately added with five percent of FA and encapsulated FA at the same moles. The characteristics of the cosmetic products were evaluated by both visual observation and equipment. Neither cream showed a phase separation when centrifuged at 9000 rpm for 30 min. The formulations were measured for viscosity (Brookfield DV-II + pro viscometer) using the RV/HA/HB-4 spindle with a speed of 50 rpm at room temperature.

### 3.11. Ethical Aspects

The protocol was approved by the Ethics Committee of Mae Fah Luang University. The certificate of analysis (COA) and protocol numbers were 168/2020 and EC 20129–17, respectively. The research protocols were conducted in agreement with the Declaration of Helsinki on human subjects. Each volunteer signed a written informed consent before participating in the clinical study, which explained the type of study, the procedures to be followed, the general nature of the materials being tested, and any known or anticipated adverse reactions that might result from participation. All volunteers who passed the occlusive single patch test for irritation upon exposure to test cream were included in the study.

### 3.12. Subjects

The 16 subjects participating in this study were concerned with the skin tone, firmness, and wrinkles. The main inclusion criteria were male and female, aged 20 to 60 with normal skin and Fitzpatrick skin type III or IV. During the study’s duration, the skincare habits and concomitant use of other skincare products were not permitted. Subjects were selected, except for those who did not meet clinical criteria that might interfere with the test’s results. The exclusion criteria included those without chronic skin disease, sensitive and irritable skin, pregnant, routinely consumed alcohol and/or smoked, and those who had participated in the same experiment within the past month.

### 3.13. Skin Irritation Testing

Skin irritation testing was performed using a Draize model, as described by Schnuch et al., 2008 [20], using Finn chambers^®®^. Skin irritation was performed on the upper arm of volunteers, with each chamber saturated by 5% *w*/*v* of the FA cream, 5% *w*/*v* of the encapsulated FA cream, 0.5% *w*/*v* of sodium lauryl sulfate (used as a positive reaction), and deionized water (as a negative reaction), before being covered for 24 h. Subsequently, we observed the erythema and edema at 30 min and 24 h after removing the patch. Each test substance was evaluated based on the mean irritation index (M.I.I) (Table 6 and Table 7).

### 3.14. Application of Encapsulated FA/FA Creams

The volunteers were instructed to apply the test product at home under normal conditions for 4 weeks, on both the left and right upper arms, and details of how the volunteers were to use the creams was explained.

A random code was assigned to each product instead of a name to avoid bias. All volunteers received two creams (the encapsulated FA and FA creams) both of which contained 5% FA. Volunteers were instructed to apply the encapsulated FA and FA creams twice daily in the morning and before bedtime. In particular, the specific instructions for this experiment were not to use any other skincare products and avoid sunlight in relevant areas.

### 3.15. Efficacy Test

In this clinical trial, volunteers’ upper arms were washed with mild body cleanser, and then, they acclimated to room conditions for at least 30 min prior to measurements being taken. All skin measurements were conducted under controlled temperatures (20 ± 5 °C) and humidity (50 ± 10%). The skin elasticity and melanin content were measured in triplicate baselines, 2 weeks and 4 weeks after use, by the single-blind method. A skin elasticity Cutometer was used to measure skin elasticity of the arm. The Cutometer^®®^ dual MPA 580 (Courage Khaazaka electronic GmbH, Cologne, Germany) was used to measure skin elasticity by evaluating the net elasticity of the skin without viscous deformation, and it was repeated 3 times to obtain an accurate measure on the same part by pressing on the measurement probe to measure the skin. The gross elasticity of the skin (R2), the net elasticity (R5) and the portion of viscoelasticity (R7) of the skin were assessed at the baseline (week 0), at week 2, and at week 4. The melanin content of the skin was measured before and after the experiment using a Mexameter (MX18, Courage Khaazaka electronic, Cologne, Germany). After contact of the sensor with the skin surface of the selected site, the melanin content was confirmed, with the mean value calculated and compared. The treated skin was evaluated using Visioscan^®®^ VC 98, which is an instrument for performing skin images where a rectangular area of the skin surface is scanned by UVA light. The roughness (SEr), scaliness (SEsc), smoothness (SEsm), and wrinkle (SEw) of the skin’s surface were assessed at the baseline (week 0), at week 2, and at week 4.

### 3.16. Statistical Analysis

Statistical analysis was performed using GraphPad Statistical Software (GraphPad Software, Inc., Version 5, La Jolla, CA, USA). Continuous variables were expressed as mean ± standard deviation (SD). The data were expressed as means and standard deviations of the triplicate values. Statistical comparisons for each experiment were performed using the paired *t*-test with a confidence interval of 95%. A *p*-value below 0.05 was considered statistically significant. All statistical analyses were performed using IBM SPSS statistics software.

## 4. Conclusions

In this study, FA was successfully encapsulated in PRS, which was demonstrated by FTIR, SEM, XRD, and DSC. The encapsulated FA was obtained with a satisfactory yield (77%), encapsulation (73%), and loading (65%) efficacies. Encapsulation with PRS was able to increase the stability of the FA during the heat and light stress conditions, thus promising an alternative method to enable FA application. The clinical evaluation of skin lightening and anti-aging indicated that the encapsulated FA cream offers a significantly higher ability to reduce the melanin content, scaliness, and wrinkle and enhance skin firmness than the FA in the long term. In conclusion, the results of this work achieve a satisfactory encapsulation of FA in PRS for enhancing stability and can be used as a skin lightening and anti-aging ingredient in cosmetic products. To our knowledge, this study is the first to encapsulate FA using phosphorylated starch as an encapsulating polymer and perform a clinical trial on its skin lightening and anti-aging efficacies. However, a bigger and more accurate clinical study has to be performed in the future.

## Figures and Tables

**Figure 1 molecules-27-03463-f001:**
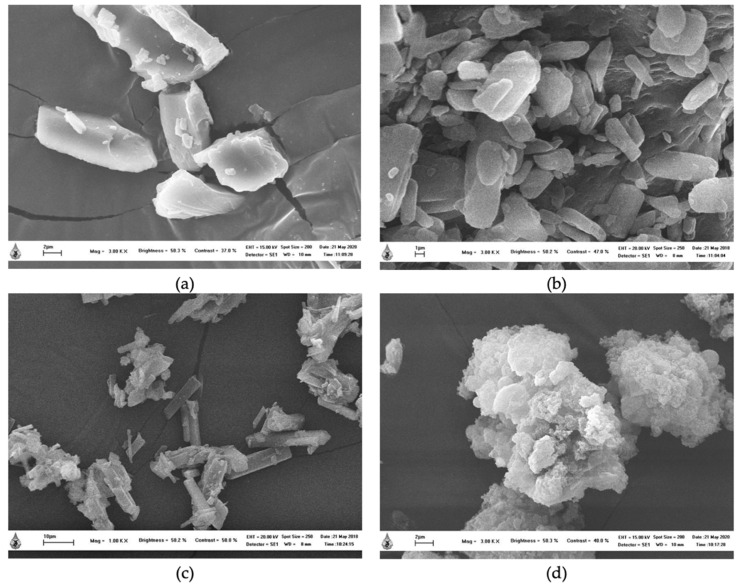
SEM images of PRS (**a**), FA (**b**), physical mixture, (**c**) and encapsulated FA (**d**).

**Figure 2 molecules-27-03463-f002:**
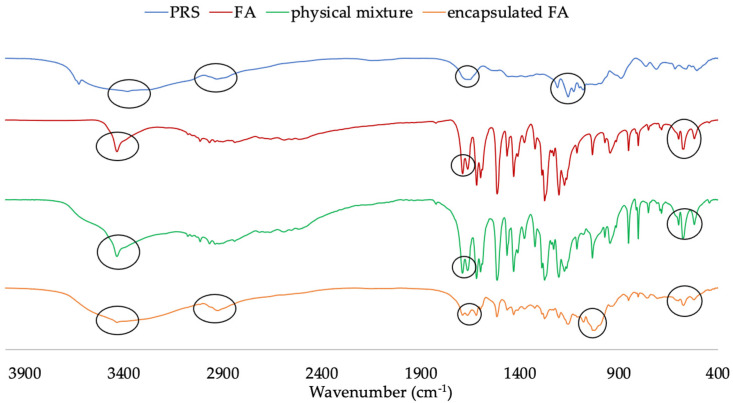
FT-IR spectra of PRS, FA, physical mixture, and encapsulated FA.

**Figure 3 molecules-27-03463-f003:**
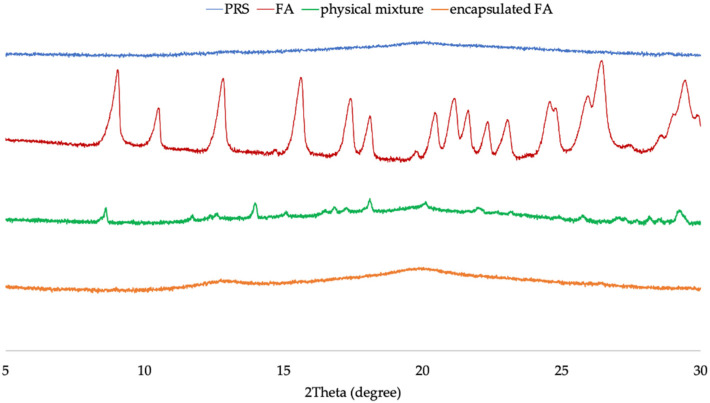
The X-ray diffractograms of PRS, FA, physical mixture, and encapsulated FA.

**Figure 4 molecules-27-03463-f004:**
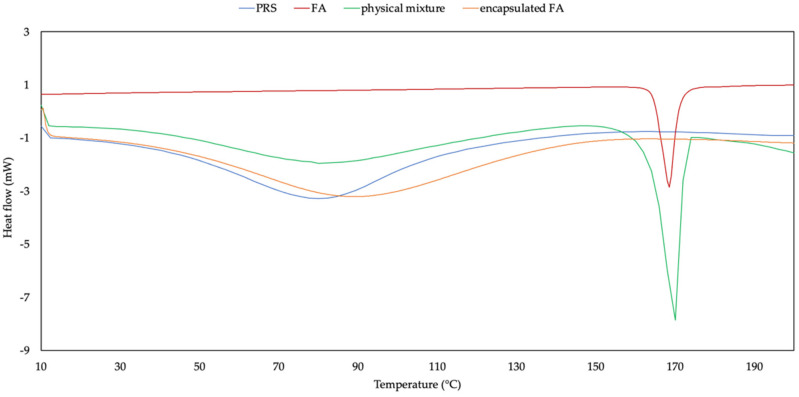
DSC curves of PRS, FA, physical mixture, and encapsulated FA.

**Figure 5 molecules-27-03463-f005:**
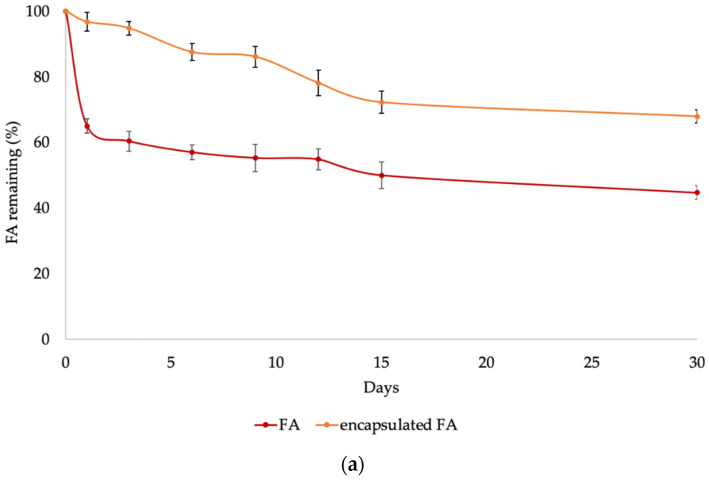
Stability of encapsulated FA and FA storage at high temperature, 80% RH (**a**) and light exposure (**b**).

**Figure 6 molecules-27-03463-f006:**
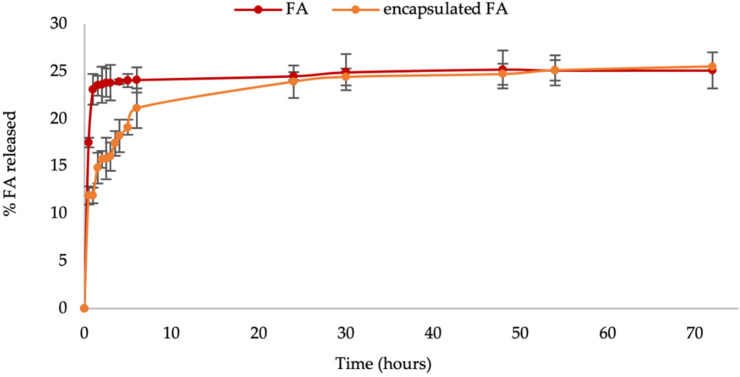
In vitro FA release from encapsulated FA compared to FA.

**Figure 7 molecules-27-03463-f007:**
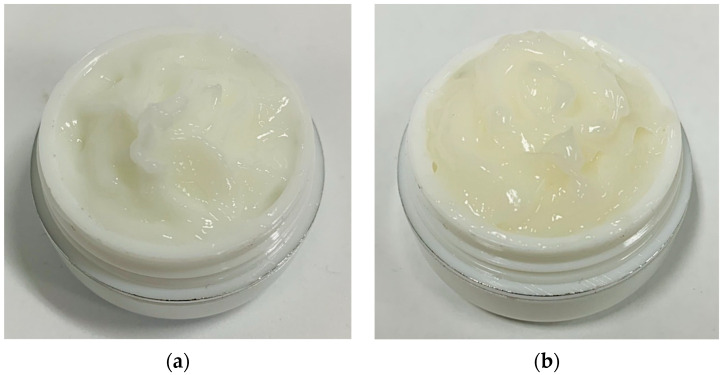
Cosmetic cream containing FA (**a**) and encapsulated FA (**b**).

**Figure 8 molecules-27-03463-f008:**
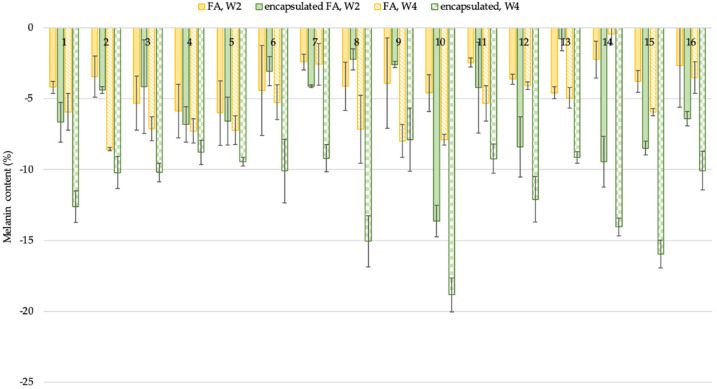
Percent of change in melanin content of each volunteer after application of FA and encapsulated FA creams for 2 and 4 weeks.

**Figure 9 molecules-27-03463-f009:**
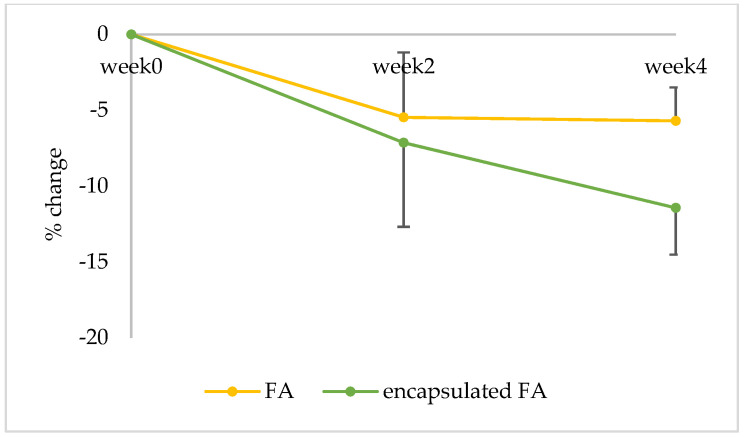
Percent of change in melanin of 16 volunteers after application of FA and encapsulated FA creams.

**Figure 10 molecules-27-03463-f010:**
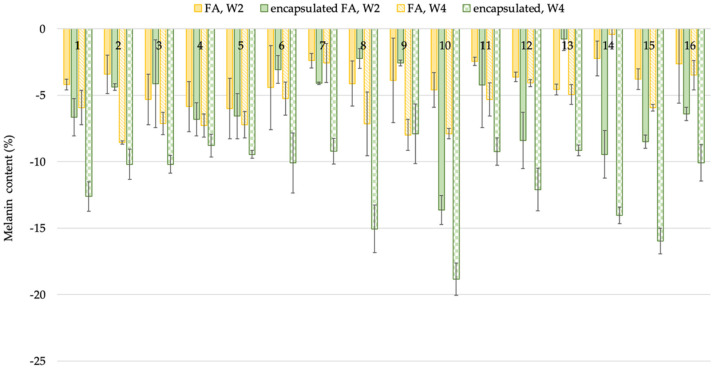
Percent of change in skin elasticity (R2) of each volunteer after application of FA and encapsulated FA creams after 2 and 4 weeks.

**Figure 11 molecules-27-03463-f011:**
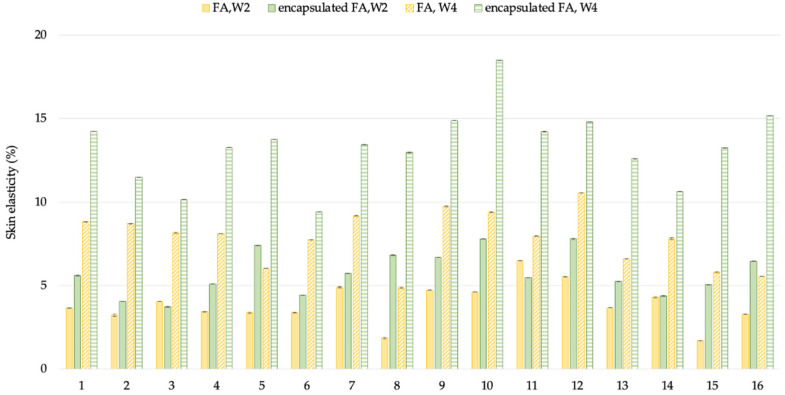
Percent of change in skin elasticity (R5) of each volunteer after application of FA and encapsulated FA creams for 2 and 4 weeks.

**Figure 12 molecules-27-03463-f012:**
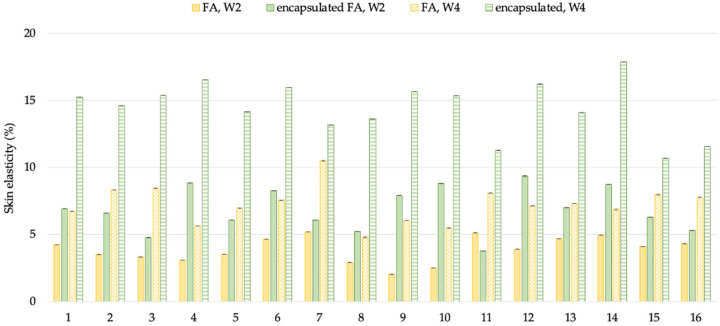
Percent of change in skin elasticity (R7) of each volunteer after application of FA and encapsulated FA creams for 2 weeks.

**Figure 13 molecules-27-03463-f013:**
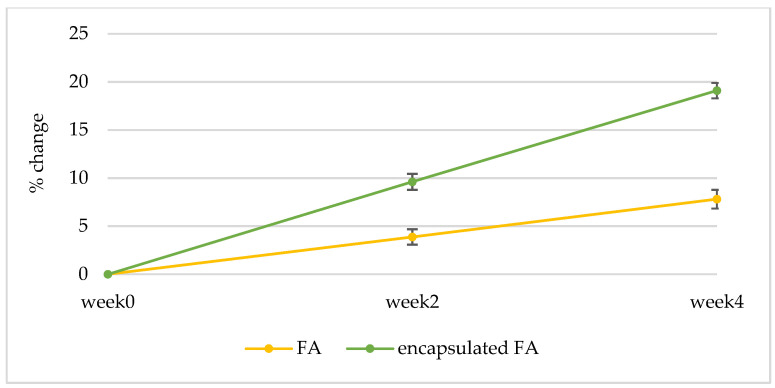
Percent of change in skin elasticity (R2) of 16 volunteers after application of FA and encapsulated FA creams.

**Figure 14 molecules-27-03463-f014:**
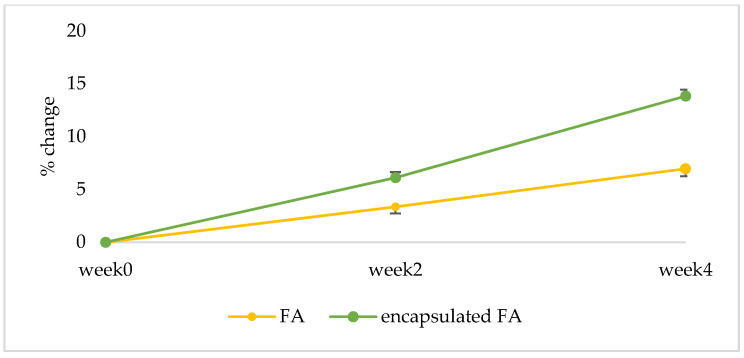
Percent of change in skin elasticity (R5) of 16 volunteers after application of FA and encapsulated FA creams.

**Figure 15 molecules-27-03463-f015:**
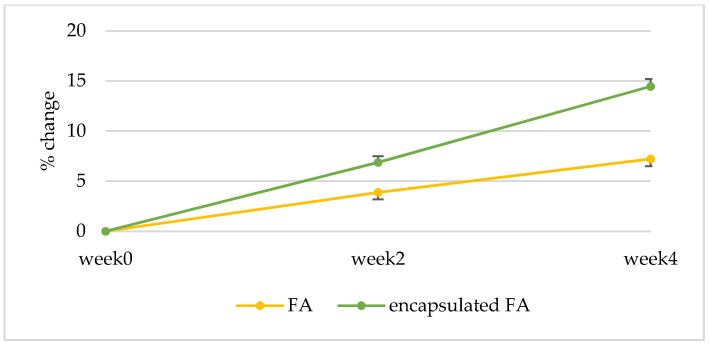
Percent of change in skin elasticity (R7) of 16 volunteers after application of FA and encapsulated FA creams.

**Figure 16 molecules-27-03463-f016:**
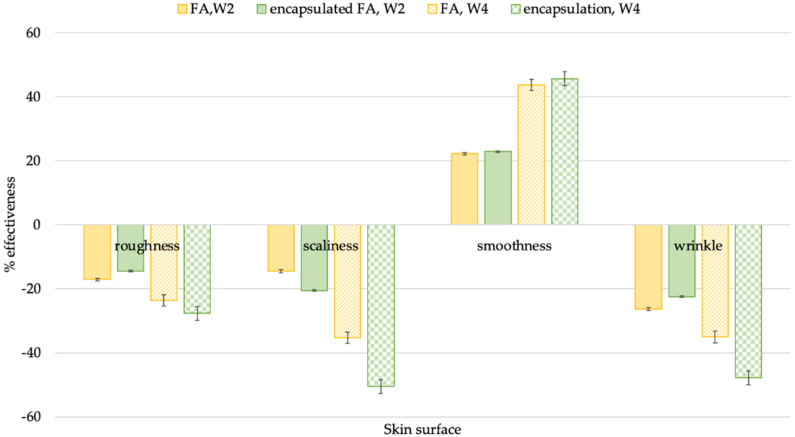
Efficiency comparison of FA and encapsulated FA creams after being applied for 2 weeks.

**Figure 17 molecules-27-03463-f017:**
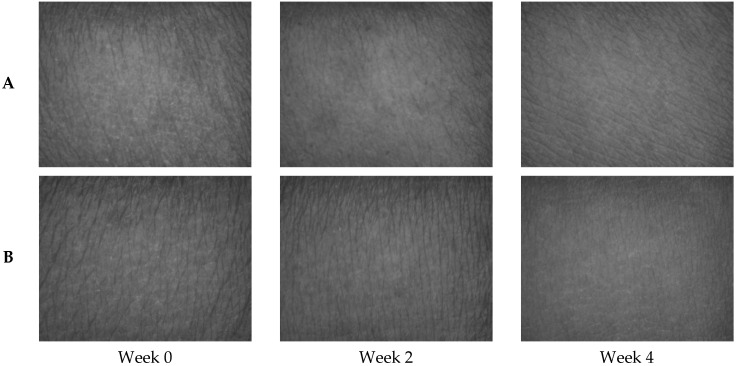
Skin comparison before and after applying FA (**A**) and encapsulated FA (**B**) cream.

**Table 1 molecules-27-03463-t001:** Key evaluating parameters of encapsulating FA with various ratios between PRS and FA.

Ratio of M_PRS_ ^a^:M_FA_ ^a^	Yield (%)	EE (%)	LE (%)
1.0:1.0	69.14 ± 0.97	24.18 ± 0.57	21.31 ± 0.65
1.0:1.5	73.20 ± 1.24	56.21 ± 0.95	38.28 ± 1.53
1.0:2.0	77.66 ± 2.25	73.10 ± 2.21	65.40 ± 2.07
1.0:2.5	77.51 ± 1.98	68.72 ± 1.99	65.07 ± 0.91
1.0:3.0	76.78 ± 2.15	66.42 ± 2.42	64.14 ± 1.34

^a^ M_PSN_ and M_FA_ presented the molar number of glucose subunit and FA applied.

**Table 2 molecules-27-03463-t002:** DSC data of PRS, FA, physical mixture, and encapsulated FA.

Sample	T_o_	T_p_	T_c_	ΔT	ΔH (J/g)
PRS	36.30	80.17	119.43	83.13	235.67
FA	174.17	177.53	180.52	6.35	92.49
Physical mixture	164.68	169.67	172.54	7.86	71.19
Encapsulated FA	37.79	88.83	141.60	103.81	262.88

T_o_, “onset” or initial temperature; T_p_, peak temperature; T_c_, “endset” or conclusion temperature; ΔT = T_c_ − T_o_ temperature range; ΔH, gelatinization enthalpy.

**Table 3 molecules-27-03463-t003:** Particle size, PDI, and zeta potential of PRS and encapsulated FA.

Sample	Z Average Size (nm)	PDI Values	Zeta Potential (mV)
PRS	71.51 ± 7.01	0.297 ± 0.007	−67.72 ± 3.18
Encapsulated FA	73.90 ± 2.67	0.394 ± 0.005	−14.85 ± 1.90

**Table 4 molecules-27-03463-t004:** The Mean irritation index and skin irritation of volunteers.

Sample	M.I.I. Value	Classification of Skin
SLS	1.12	Slight irritation
Water	0.00	No irritation
FA cream	0.00	No irritation
Encapsulated FA cream	0.00	No irritation

**Table 5 molecules-27-03463-t005:** Base cream formulation.

Phase	Ingredients	*%w/w*
A	Water	86.0
	Glycerin	5.0
	Butylene glycol	2.0
	Acrylates/Acrylamide Copolymer (and) Mineral oil (and) Polysorbate 85	1.5
B	Glyceryl Stearate SE	1.2
	Cetearyl alcohol	1.5
	Glyceryl Stearate (and) PEG-100 Stearate	1.5
C	Phenoxyethanol	0.8

**Table 6 molecules-27-03463-t006:** Scores about erythema, oedema, or other skin irritations.

Score	Clinical Description
0	No erythema
1	Light erythema (hardly visible)
2	Clearly visible erythema
3	Moderate erythema
4	Serious erythema (dark red with possible formation of light scars)
0	No oedema
1	Light oedema (hardly visible)
2	Light oedema
3	Moderate oedema (about 1 mm raised skin)
4	Strong oedema (extended swelling even beyond the application area)

**Table 7 molecules-27-03463-t007:** Classification of M.I.I. (according to the amended Draize classification).

M.I.I.	Classification
<0.5	Non-irritation
0.5 to 2.0	Slight sirritation
2.0 to 5.0	Moderate irritation
5.0 to 8.0	Strong irritation

## Data Availability

All the data are available in the manuscript.

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
