# Peer review of "Stability and Anti-Aging of Encapsulated Ferulic Acid in Phosphorylated Rice Starch"

_molecules, 2022, doi:10.3390/molecules27113463_

Round 1

Reviewer 1 Report

Recommendation: major revisions needed.

This study aimed to enhance the stability of FA by encapsulating in phosphorylated rice starch (PRS) and evaluate its effect on improving human skin. Whether ferulic acid (FA) is encapsulated in phosphorylated rice starch (PRS) remains to be further verified. Therefore, I suggest that this paper be published in the Molecules after a major revision. The specific revisions are as follows:

  1. The characterization patterns of FTIR, XRD and DSC in this paper have not examined the physical mixture, so it is difficult to explain whether ferulic acid (FA) is successfully encapsulated in phosphorylated rice starch (PRS). It is possible that the authors prepared solid dispersions or co-amorphous systems of ferulic acid (FA) and phosphorylated rice starch (PRS).
  2. In the infrared spectrum, the authors should mark the changes in the main functional groups of ferulic acid (FA) and phosphorylated rice starch (PRS).
  3. For crystalline materials, there is a fixed melting point, but for amorphous materials, there is no fixed melting point. The thermodynamic phenomenon that should appear for amorphous materials is that the glass transition occurs first, the exothermic peak appears after the transformation, and finally, the endothermic peak appears after the transformation. Therefore, the thermodynamic phenomena of amorphous materials should be explained clearly.
  4. In the stability test of the influencing factors, the author only carried out high temperature and light experiments. Why did not do a high humidity test?
  5. For Figure 6 the standard deviation should be displayed.

Author Response

Dear Editors:

Thank you for the opportunity to revise our manuscript, Stability and anti-aging of encapsulated ferulic acid in phosphorylated rice starch. We appreciate the careful review and constructive suggestions. It is our belief that the manuscript is substantially improved after making the suggested edits.

Following this letter are the editor and reviewer comments with our responses in the table, including how and where the text was modified. The revision has been developed in consultation with all coauthors, and each author has given approval to the final form of this reversion.

We believe have resulted in an improved revised manuscript, which you will find uploaded alongside this document. The manuscript has been revised to address the reviewer comments, which are appended alongside our responses to this letter.

We very much hope the revised manuscript is accepted for publication in Molecules Journal.

Sincerely yours,

Nisakorn Saewan, Ph.D.

Reviewer 2 Report

The authors reported the use of ferulic acid, free or encapsulated on PRS to enhance the stability of this anti-aging compound. The data is well supported by the characterization performed on volunteers.

I have some comments/questions:

  • Line 124: The term gelatinization is not very usual in the description of DSC results.
  • DSC results: Even if a compound is not crystalline, it still needs to have a melting point. An extended discussion regarding this topic should be included.
  • Section 2.6.: The results regarding polydispersity are missing. Authors should include PDI values, given that is an important parameter of the characterization of the particles produced. Moreover, it does not make sense to measure the size of the FA alone.
  • Figure 5. The standard deviation on the graphs are missing.
  • Figure 6. The graph of the release should be in %.
  • Section 2.9. Authors claim that the creams “had appreciable texture, spreadability, thickness and smoothness.” How were these parameters measured? Did they use a Texture analyzer to obtain these conclusions?
  • Line 220: the meaning of M.I.I. should be included here, since the materials and methods section are at the end of the manuscript.
  • Figure 8 +9. Both these graphs should be combined in 1 figure, for a better and easier comparison of the results at 2 weeks and 4 weeks. The same should be done for Figure 11 + 12; Figure 14 + 15; Figure 17 + 18.
  • Graphs of the figures 20 + 21 could be combine in one.
  • Figure 22 and 23 could be also lined together to observe the differences between free and encapsulated FA on the creams.
  • I don’t understand how the LE was calculated. By the formula, I don’t understand what is the difference between weight of FA in encapsulated and Total weight of encapsulated FA…

Author Response

(The authors gave the same response as above.)

Reviewer 3 Report

It is a well written and illustrated article. A lot of  experiments have been performed and are  sufficiently approved. There are some improvements, especially in design of clinical tests,  which should have been done and wound enhance the scientific research and interest.                             I prefer the Materials and methods to be written before the results, since many questions of reader may have been prevented. Furthermore, It would be beneficial to separate the Discussion from the Results illustration, which wound make the research article more attractive and interesting.

Line 30: Add the antimicrobial property of Ferulic acid, since it is officially referred in CosIng EU legislation.

Line 159: You have to explain why you have not used in stability tests the Humidity parameter or if missed which is the accurate Humidity ;

Line 165: You have to explain or justify why you have not used in stability tests the UV light and you have chosen the Fluorescent light  or parameter ;

Line 208: Replace the thickness by viscosity and add the consistency with the rest characteristics.

3.10 Formulation:  You have to write which is the pH of finished product and refer the pH modifier if used.

Table 5. : Replace the trade names of Novermer EC-1, GMS SE and Lexmul 561 with INCI names or chemical names.

3.14: It is not referred how volunteers were used the creams together, if I can understund well. You have to clarify the way (half face, left-right hand / arm). It is very important, especially because you have not performed a double blind clinical study.

Line 494: Replace the word irradiated by scanned.

Finally, you have to refer  that a bigger and accurate clinical study has to be performed in the future.

Author Response

(The authors gave the same response as above.)

Round 2

Reviewer 1 Report

Recommended to accept in present form